DATA RELEASE

# Digitizing the Culicidae collection of Naturalis Biodiversity Center, with a special focus on the former Bonne-Wepster subcollection

Pasquale Ciliberti[1,*], Astrid Roquas[1], Becky Desjardins[1], Bibiche Berkholst[1], Frank Loggen[1], Menno Hooft[1], Gideon Gijswijt[1] and Dick de Graaff[1]

1 Naturalis Biodiversity Center, Darwinweg 2, 2333 CC, Leiden, The Netherlands

## ABSTRACT

Natural history collections contain a wealth of information on species diversity, distribution and ecology. However, due to historical and practical constraints, this valuable information is not always available to researchers. Our project aimed at unlocking data handwritten in notebooks owned by Johanna Bonne-Wepster, a Culicidae researcher. These handwritten notes refer to specimens labeled with a number only. The notebooks were scanned and entered into a Google spreadsheet. The specimens were provided with a unique identifier, labeled with the information from the notebooks and the data exported to the Global Biodiversity Information Facility. In addition, the type specimens were photographed. Besides Johanna Bonne-Wepster's collection, mosquitoes from the former Rijksmuseum van Natuurlijk Historie collection and the former Zoölogisch Museum Amsterdam Nederland collection were digitized. All specimens are now housed at the Naturalis Biodiversity Center museum in Leiden. This paper describes the efforts to mobilize this data and the problems we encountered.

**Subjects** Ecology, Biodiversity, Taxonomy

**Submitted:** 27 February 2023

\* Corresponding author. E-mail: pasquale.ciliberti@naturalis.nl

Preprint submitted at https://doi.org/10.5281/zenodo.7969565

Included in the series: *Vectors of human disease series* (https://doi.org/10.46471/GIGABYTE_SERIES_0002)

## DATA DESCRIPTION

We describe a dataset of 55,706 records of mosquitoes, mainly collected and observed in the former Dutch colonies of Indonesia and Suriname. The dataset includes primarily mosquitoes identified and owned by Johanna Bonne-Wepster. This collection is now maintained by the Naturalis Biodiversity Center (NBC) in Leiden, The Netherlands.

The Bonne-Wepster mosquitoes were collected by staff from the Centraal Militair Geneeskundig Laboratorium (CMGL), by fellow mosquito taxonomists and by Bonne-Wepster herself. In addition, many mosquitoes were collected by the South East Asian Mosquito Project (SEAMP), an international collaboration of the American and Dutch armies. The mosquitoes collected within the SEAMP project were maintained and curated at the Instituut voor Tropische Hygiëne (ITH) in Amsterdam, the Netherlands. The Bonne-Wepster collection was transferred during the seventies to the Rijksmuseum van Natuurlijk Historie in Leiden (RMNH).

The information about the specimens, including locality data and identifications, was stored in eight field books, five from CMGL and three from ITH. These field books used the same numerical order, so a given registration number could refer to two separate

**Table 1.** Composition of the specimens digitized during this project.

| Former owner | Structure of collection | Number of specimens | Historical notes |
|---|---|---|---|
| Bonne-Wepster | Formed by former CMGL collection and former ITH collection and her own sampled specimens. | 52,102 of which 40,705 records are observation only for the missing CMGL specimens. | Since the 70's of last century part of the RMNH collection and since 2011 part of the NBC collection |
| RMNH | | 2,388 | Since 2011 part of the NBC collection |
| ZMAN | | 1,216 | Since 2011 part of the NBC collection |

specimens. In the collection, the specimens were labeled only with a number. Due to the age of the field books, we feared losing the information they contained. Therefore, the field books were scanned, specimens were provided with labels and the data was entered in a database.

Before we started our project, we found that some mosquitoes with a Bonne-Wepster number were already provided with locality and species labels. Most CMGL mosquitoes are missing; therefore, these specimens could only be digitized as observation records (40,705). In addition, 1,216 mosquitoes from the former Zoölogisch Museum Amsterdam Nederland (ZMAN) collection and 2,388 mosquitoes from the former RMNH collection were digitized as well. All specimens belong to the NBC collection (Table 1).

## CONTEXT

Natural history collections are a rich data source that can be used for scientific research, education and the general public [1]. Traditionally, specimens were collected for private use in 'cabinets of curiosity' and subsequently adopted by museums, where they became the focal point of taxonomic research [2].

In recent years, natural history collections have embraced other fields of the biological sciences besides taxonomy. Despite sampling biases, natural history collections can be used to model current and past distributions [3], build molecular libraries [4], analyze biodiversity (i.e., for conservation purposes) [5], assess hybridization and speciation events [6], and predict the reemergence of diseases [7].

The natural history collection of NBC contains about 42 million objects sampled over the past 200 years [8]. A small part of this collection is formed by mosquitoes. Mosquitoes are among the most feared insects for their vector role in transmitting a wide array of pathogens, the most notorious being protozoans of the genus *Plasmodium*, the causative agents of malaria [9]. According to the World Health Organization, 619,000 people died of malaria in 2021 [10].

Most of the mosquito specimens now housed at NBC were owned and identified by Johanna Bonne-Wepster. The collection was used to study the morphological features that characterize mosquito species. The specimens were provided with labels having a handwritten number pinned below them (Figure 1). These numbers were linked to information contained in eight different field books (Figure 2). The data was organized in columns indicating the species name, a description of the collection place, the collector and, sometimes, short taxonomical notes (Figure 3). As the field books of Johanna Bonne-Wepster are very old, we feared losing the data within and, consequently, the value of the specimens. Indeed, as beautifully stated by Lane [11], a specimen separated from its label has no scientific value:



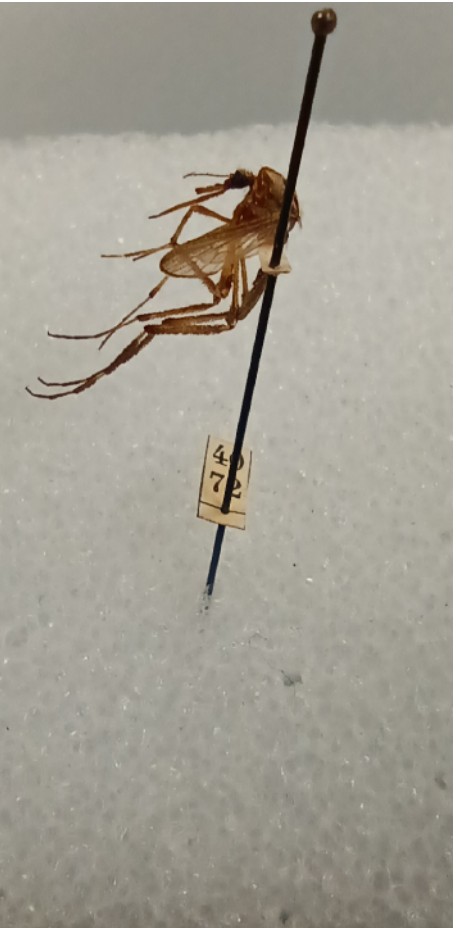

**Figure 1.** Example of a specimen labeled with a number only.

*'Together, a preserved organism and its label are a scientific specimen that has great intrinsic value. Separately, the label is a piece of paper with meaningless inscriptions upon it, and the plant, spider, microbe, mushroom, or bird, though carefully preserved, is just so much dead organic matter.'*

*(Lane, 1996: 536)*

Hence, we strongly felt the need to 'rescue' the information in the field books and link this information to the specimens.

## JOHANNA BONNE-WEPSTER

Johanna Wepster was born in 1892 in The Hague. She was trained as a teacher. She married Cornelis Bonne, who was two years older and trained as a physician, specializing as a parasitologist with an interest in tropical medicine. Together, they went to the Dutch overseas colonies.

Bonne worked in Suriname first. Then, in 1927, the couple headed for the Dutch East Indies. He studied mosquitoes and especially their role in spreading pathogens. Mrs. Bonne-Wepster started collecting mosquitoes at a large scale. In both Suriname and the

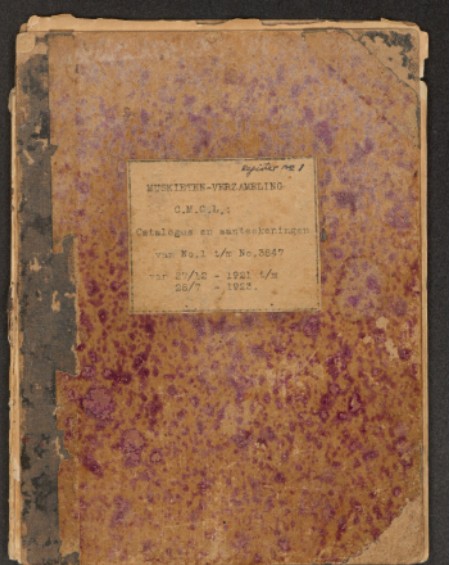

**Figure 2.** Field book.

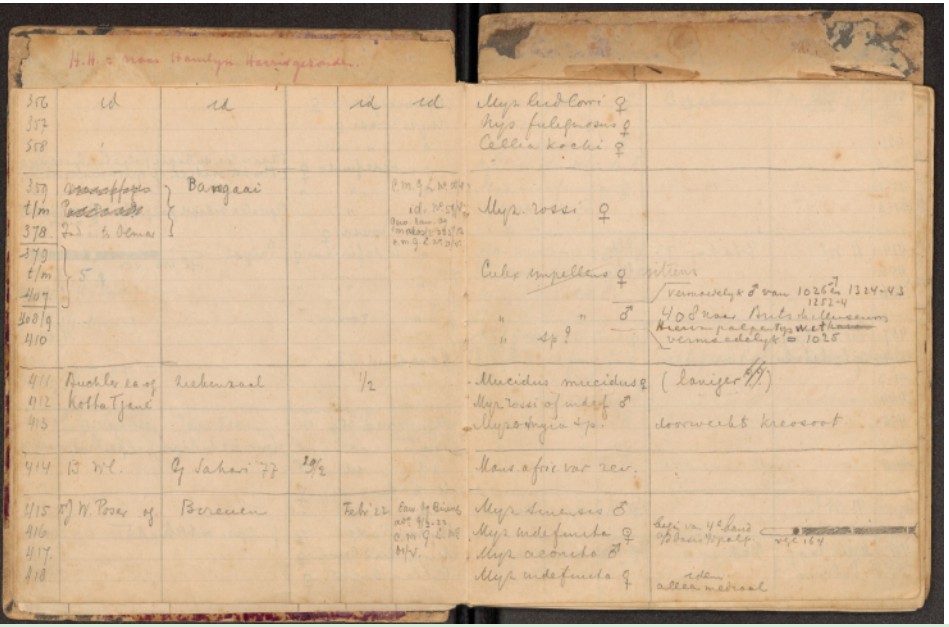

**Figure 3.** Example of the data contained in a field book.

Dutch East Indies, she conducted thorough investigations about this family of blood-sucking insects. Her main goal was to give non-taxonomists the means to recognize vector species.

Mrs. Bonne-Wepster wrote and co-wrote several publications on the taxonomy of Culicidae [12–17]. During the twenties and thirties, she collected thousands of specimens: many adults, but also eggs and larvae.

In addition, the couple contributed to the SEAMP, an international collaboration between American and Dutch militaries. This important research project lasted until after the



Second World War. After their return to The Netherlands in 1948, Cornelis Bonne died. Johanna continued researching Culicidae, primarily at the University of Amsterdam. Even though she was not academically trained, her enormous contribution to the field of Culicidae taxonomy did not go unnoticed: she received an honorary doctorate from the University of Amsterdam in 1951.

In the seventies, her collection was transferred to the RMNH in Leiden (Table 1). Initially, the museum obtained only the sampled specimens, not the field books with detailed field notes. The field books appeared to be lost until a curator of the RMNH visited the elder Mrs. Bonne-Wepster and retrieved them. Mrs. Bonne-Wepster passed away in 1978.

## MATERIAL AND METHODS

### Purpose

The main purpose of the project was to mobilize the data contained in the field books of Bonne-Wepster and corroborate it with the associated specimens. As we proceeded with the project, mosquitoes of the former ZMAN collection and the RMNH collection were digitized as well.

### Sampling description

All specimens were carefully checked, investigated and provided with unique registration numbers (Figure 4). This project produced 55,706 records. Among them, 52,102 records originated from the former Bonne-Wepster collection, including 40,705 missing CMGL specimens that were digitized as observation records. Additionally, 2,388 records pertain to mosquitoes from the former RMNH Culicidae collection, and 1,216 records refer to specimens from the former ZMAN Culicidae collection (Table 1).

### Process

(1)  First, we entered the data of the field books in a Google spreadsheet.

(2)  Next, the field books were scanned and stored according to the Naturalis Archive protocol.

(3)  The entered records were georeferenced using the Point Radius method, as described by Wieczorek *et al.* [18] (see below: Coordinates).

(4)  Unique registration numbers with a corresponding QR code were added to the specimens. When the specimens were only labeled with a Bonne-Wepster number, additional labels were printed and added. Those labels were (a) a locality label, (b) a species name label and (c) an 'ex. coll. Bonne-Wepster' label (Figure 4).

(5)  The collection contained about 50 (holo)types, which were photographed (Figure 5) using a Zeiss Discovery.V12 modular stereo microscope equipped with an AxioCam Mrc 5 camera. Depending on the state of the specimen, photographs were taken of the habitus (dorsal and lateral views). A scale bar in micrometers was added to the photos. The labels accompanying the (holo)types were photographed using a Nikon camera D600 equipped with an AF Micro-Nikkor 60 mm f/2.8D lens. An overhead camera setup was used to photograph the labels from above.

(6)  The Google spreadsheet was converted into a standardized sheet format that could be imported into the NBC database. Table 2 presents an overview of the specimen-specific information included in this sheet.



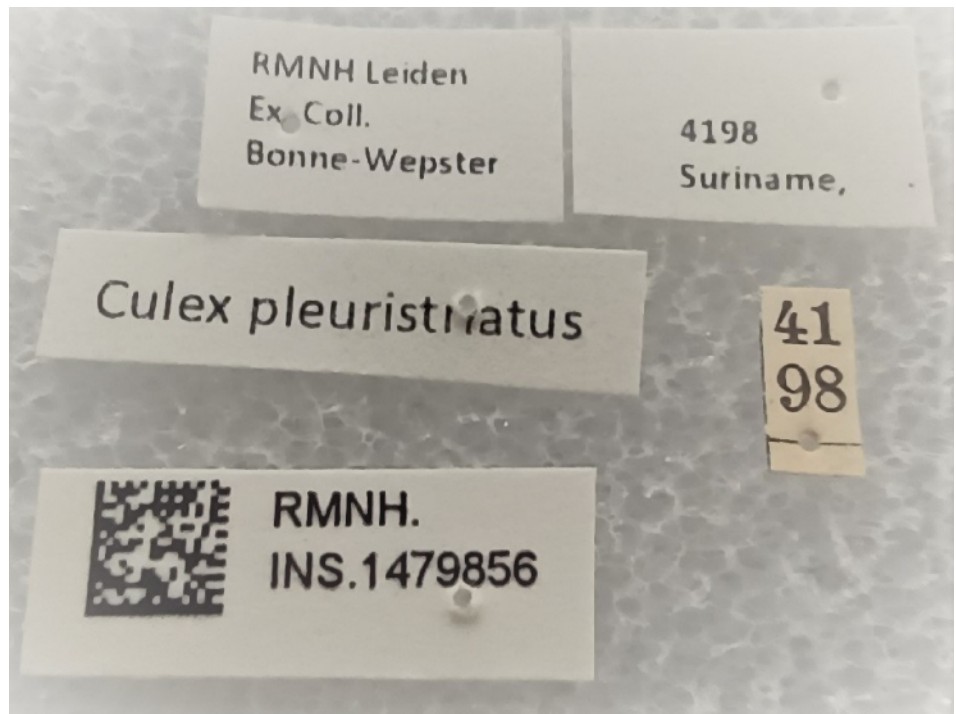

**Figure 4.** Examples of labels added during this project when only a number was pinned under the specimen. Note the unique registration number with a QR code added to all specimens.

## Coordinates

To indicate the coordinates, we used Google Earth and the Georeferencing calculator [19]. We used the point radius method technique. The point radius method delineates a locality via a pair of points and a distance, with the distance being a radius describing a circle around the points [18].

The field data of the Bonne-Wepster collection did not contain any coordinates. Georeferencing an 'old' collection is challenging. The locality descriptions often lacked specificity. When the locality name was not specific enough (for instance, The Lawa River, Suriname) or unknown locality names were used, we did not assign any coordinates. The maximum length of the radius was set at 100 km.

Since the gathering area of this collection was widespread – from Sydney, Australia to Whitehorse, Canada, and from Transvaal, South Africa to Pampanga, Philippines – an indication of the minimum and maximum latitude and longitude was considered pointless.

## Data validation

The field books were handwritten, so interpreting the locality and species names could be challenging. Some mosquitoes with a Bonne-Wepster number already had a locality and identification label. We always checked if the locality on the label coincided with that in the field book. When we found such a mismatch, we reported it in the section General Remarks of the Google spreadsheet. The same was done when a mismatch concerned the species name, and this mismatch was annotated in the Name comments section of the Google spreadsheet.

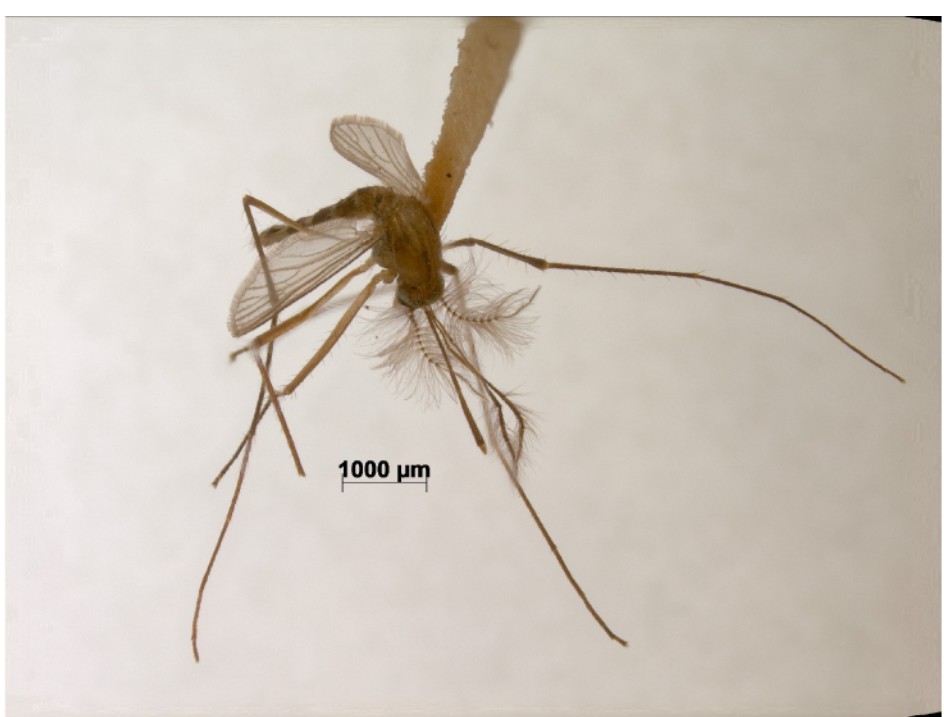

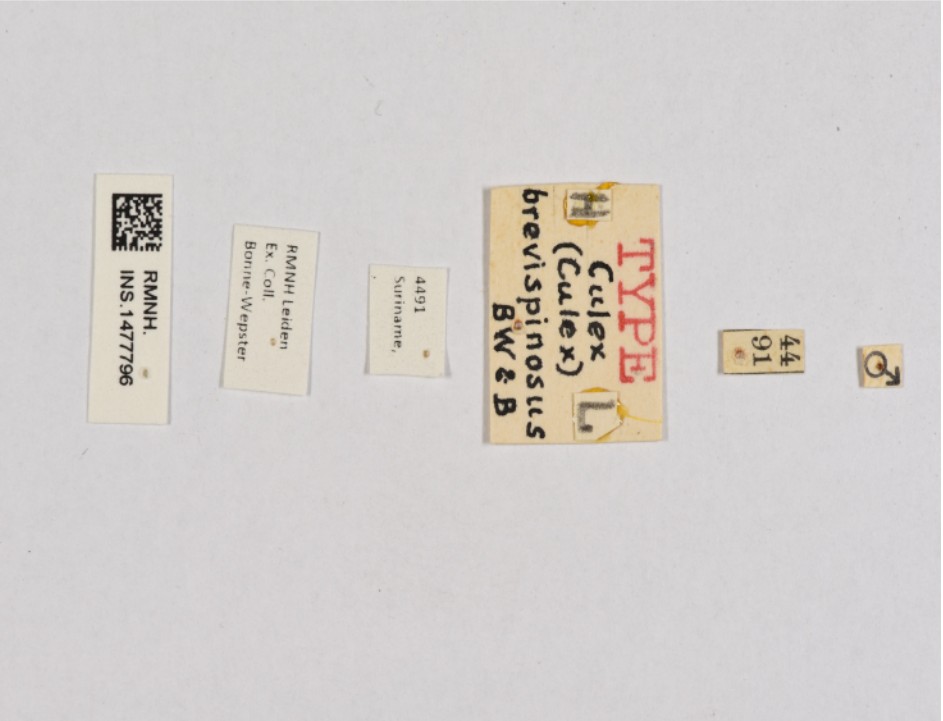

**Figure 5.** Above: example of a type specimen (*Culex brevispinosus* Bonne-Wepster & Bonne, 1920). Below: labels of the type specimen *Culex brevispinosus*.

| Table 2. | Fields used in the Google spreadsheet and their descriptions. |
|---|---|
| Number type | Indicates the origin of a certain number. This translates to 'old registration number' for all records. |
| Number value | Value of the Bonne-Wepster old specimen number, RMNH old specimen number and ZMAN old specimen number. |
| Register | Differentiates between the eight different field books, for the ITH as well as the CMGL collections. |
| RMNH-reg.nr. | Unique registration number added during this project. |
| BE.-nr. | Registration number of the current storage unit, which was added during this project. |
| Present_in_coll. | Presence of the specimen: 'Yes' if present and No/Missing if the specimen was not found. |
| General remarks | (1) Verbatim information from the field books not suited for registration in other fields. (2) Comments from collaborators handling the specimen, such as 'Genitals present on pin'. |
| Collector | Collector of the specimen, verbatim. |
| Datum tekstueel | Collection date, verbatim. |
| Coll. date start | Collection start date: dd/mm/yyyy. |
| Coll. date end | Collection end date, if collection date covers a range of dates: dd/mm/yyyy. |
| Historical owner | Indicates if the specimen was part of a significant historical collection (e.g., J. Bonne-Wepster). |
| Sex | Sex of the specimen, if known: male/female. |
| Phase or stage | Phase or stage of the specimen: primarily adult or larva. |
| Country StateProvince Island Locality | Geographic information on the gathering site of the specimen. If necessary, historic 'Country', 'State/Province', 'Island' and other location names are converted into current names. |
| Full locality text | All geographic information on the gathering site of the specimen, verbatim. |
| Altitude | Altitude in meters. |
| Order Family Genus Species Variety rank Variety Author | Taxonomic identification of the specimen. |
| Certainty | Certainty of the identification of the specimen. 'Uncertain' if the identification was doubtful. |
| Name comments | Additional comments regarding the identification of the specimen, such as specific information based on which the specimen was identified. Also, any discrepancies between the identification from the field books and the one on the pin were registered in this field. |
| TYPE status | If applicable, type-information of the specimen, verbatim. |
| Determinator | Name of the identifier of the specimen, verbatim. |
| Det. date | Date of the determination template: dd/mm/yyyy. |
| Latitude Longitude | Georeference information: coordinates. |
| Uncertainty | Margin of error of the coordinates (in meters) |
| Method | Point radius method (see Coordinates) |
| Remarks_coordinates | Additional comments regarding georeference information. E.g., remarks when a locality was not found or the margin of error proved was too high (>100 km). |

To preserve the historical character of the collection as much as possible, we decided not to synonymize the species names. Therefore, the dataset is provided with the names as originally given by Bonne-Wepster. However, spelling mistakes were corrected using the taxonomic checklist Culicipedia [20]. The same checklist was used to interpret genus and species names, when they were abbreviated in the field books. This ensured that the names imported into the NBC database and ultimately exported to the Global Biodiversity Information Facility (GBIF) platform were error-free.

We never removed any labels under a specimen, and it was difficult to establish how discrepancies came about. Possibly, a mosquito expert later redetermined some specimens, or mistakes were made during the previous labeling process, but this is merely speculation.

**Table 3.** Genera included in our mosquito database.

| Rank | Scientific name | Common name |
|---|---|---|
| Kingdom | Animalia | Animals |
| Phylum | Arthropoda | Arthropods |
| Class | Insecta | Insects |
| Order | Diptera | Two-winged insects |
| Family | Culicidae | Mosquitoes |
| Genus | Aedeomyia, Aedes, Aedimorphus, Anopheles, Armigeres, Banksinella, Bironella, Cancraedes, Cellia, Coquillettidia, Culex, Culiciomyia, Culiseta, Deinocerites, Dendromyia, Desvoidya, Ficalbia, Finlaya, Goeldia, Haemagogus, Harpagomyia, Heizmannia, Hodgesia, Howardina, Joblotia, Leicesteria, Leucomyia, Limatus, Lophoceratomyia, Lorrainea, Lutzia, Malaya, Mansonia, Mansonioides, Megarhinus, Miamyia, Mimomyia, Mochthogenes, Mucidus, Myzomyia, Neomelanoconion, Neomyzomyia, Nyssorhynchus, Ochlerotatus, Orthopodomyia, Pardomyia, Pseudoskusea, Psorophora, Rachionotomya, Rachiosura, Runchomyia, Sabethes, Sabethoides, Skusea, Stegomyia, Stethomyia, Taeniorhynchus, Theobaldia, Topomyia, Toxorhynchites, Trichoprosopon, Tripteroides, Uranotaenia, Verrallina, Wyeomyia | |

Sometimes, the same number could refer to two specimens (e.g., no. 234, one specimen from the former CMGL collection or one from the former ITH collection). This could be a problem in establishing the collecting event. Working intensively with the specimens, we noticed a difference in how the two subcollections were labeled. The ITH collection had numbers written vertically (Figure 6), while the CMGL collection had numbers written horizontally (Figure 6).

## GEOGRAPHIC COVERAGE

The specimens were collected in the following countries:

Indonesia, India, Iran, Sri Lanka, Yemen, Philippines, China, Thailand, Suriname, Brazil, Panama, Guatemala, Peru, Costa Rica, Cuba, Jamaica, Nicaragua, Bahamas, Trinidad & Tobago, Dominican Republic, El Salvador, Ecuador, Guadalupe, Virgin Islands, France, Belize, Puerto Rico, Colombia, Canada, Mexico, U.S.A., Netherlands, Belgium, Russia, Croatia, Spain, North Macedonia, Bulgaria, Romania, Israel, Palestine, Italy, France, Portugal, Australia, Papua New Guinea, Congo, South Africa, The Gambia, Republic of Guinea, and Mozambique.

The vast majority of the specimens were collected in Indonesia. Suriname ranks second in the number of collected specimens.

## TAXONOMIC COVERAGE

The genera included in the database are listed in Table 3.

## RE-USE POTENTIAL

The data made available through this project is valuable because it describes historical and recent records of mosquitoes and can be used in different research areas, including estimates of spatial distribution, modeling of current and future distributions through the Ecological Niche Modeling tool, systematics and vector control programs.

The species names attached to the records are the original ones and were not synonymized.

## DATA AVAILABILITY

The dataset is available in the GBIF repository [21]. The original field books of Johanna Bonne-Wepster have all been digitized and are available in the Naturalis digital

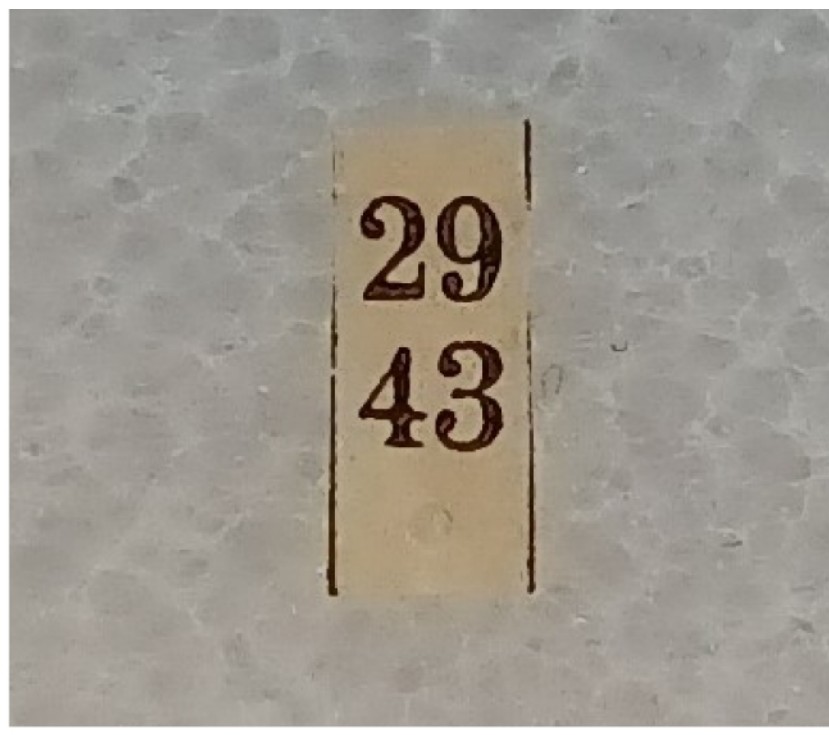

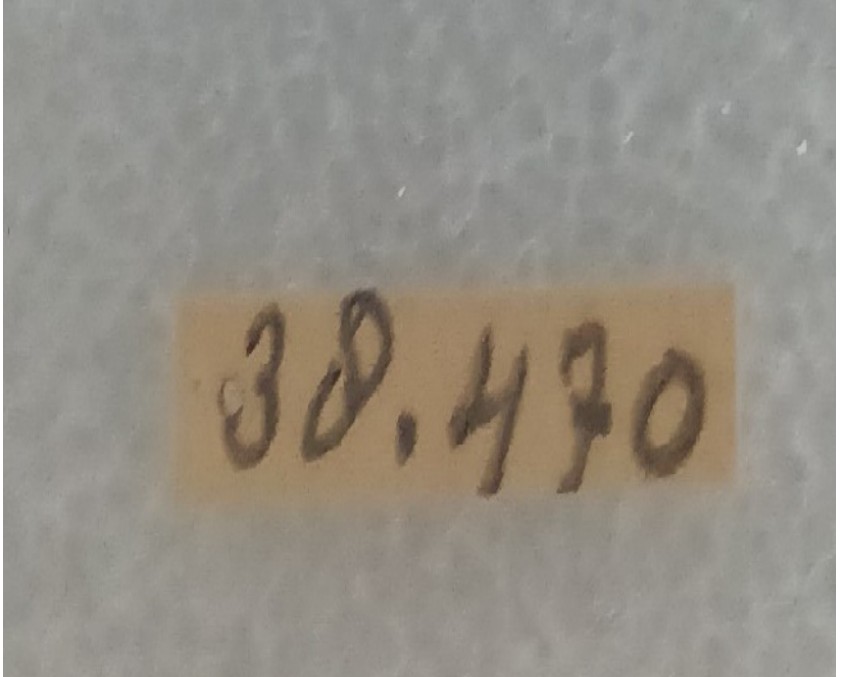

**Figure 6.** Above: example of a typical ITH label number. Below: example of a typical CMGL label number.

archives [22]. Links to the specific handles and snapshots of the archives are also available in GigaDB [23].

## EDITOR'S NOTE

This paper is part of a series of Data Release articles working with GBIF and supported by TDR, the Special Programme for Research and Training in Tropical Diseases hosted at the World Health Organization [24].

## LIST OF ABBREVIATIONS

CMGL: Centraal Militair Geneeskundig Laboratorium; GBIF: Global Biodiversity Information Facility; ITH: Instituut voor Tropische Hygiëne; NBC: Naturalis Biodiversity Center; RMNH: Rijksmuseum van Natuurlijk Historie; SEAMP: South East Asian Mosquito Project; ZMAN: Zoölogisch Museum Amsterdam Nederland.

## DECLARATIONS

### Ethical approval

Not applicable.

### Consent for publication

Not applicable.

### Competing Interests

The author(s) declare they have no competing interests.

### Authors' contributions

PC assigned coordinates, digitized specimens, took pictures of the types, quality control of the spreadsheet, wrote the paper, applied for funding. AR entered the field books data in the spreadsheet, digitized specimens, wrote the paper. BD wrote the paper. BB digitized specimens, took pictures of the types, quality control of the spreadsheet, wrote the paper. FL assigned coordinates, digitized specimens, curated and prepared the spreadsheet, quality control of the spreadsheet, wrote the paper. MH scanned the field books. GG curated and imported the spreadsheet into the NBC database. DdG assigned coordinates, digitized specimens, quality control of the spreadsheet, wrote the paper, applied for funding.

### Funding

This work was co-funded by NLBIF, nlbif2020.007.

### Acknowledgements

We are very thankful to NLBIF for funding the project. Many thanks to Caroline Pepermans for helping coordinate the team at the beginning of the project and to Jeroen Creuwels for his suggestions on how to assign coordinates and his precious help in generating the DOI. Thanks to Maarten Koster for his valuable help in entering the data in the Google spreadsheet on a voluntary basis at the very beginning of the project. We are grateful to Hendrik van Ede, Mark Doeland and Tamara de Reus for their help assigning the coordinates.



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
