## [Editor Report]

Comments to the Author The renowned entomologist Johanna Bonne-Wepster over from 1917 to 1961 collected huge collections of mosquito records mostly in the former Dutch colonies of Indonesia and Suriname. Efforts have been taken to digitise her original field books, mobilizing data contained in them and corroborating this with the associated specimens. These archival texts and 55,706 records of mosquitoes are for the first time being made public via the GBIF repository. The data made available through this project being particularly valuable because it describes historical and recent records of mosquitoes and can be used in different areas of research including estimates of spatial distribution, modeling current and future distributions through Ecological Niche Modeling, and for vector control programs.

---

## [Reviewer Report]

Upload additional filesDRR-202302-03/form/DRR-202302-03_Data-Review-MAT (1).pdfReviewer name and names of any other individual's who aided in reviewer Mary Ann TuliDo you understand and agree to our policy of having open and named reviews, and having your review included with the published papers. (If no, please inform the editor that you cannot review this manuscript.)YesIs the language of sufficient quality?YesPlease add additional comments on language quality to clarify if needed
n/aAre all data available and do they match the descriptions in the paper? YesAdditional CommentsData has been made available via https://doi.org/10.15468/dl.crvd8z
I recommend this DOI be included in the manuscript.Are the data and metadata consistent with relevant minimum information or reporting standards? See GigaDB checklists for examples <a href="http://gigadb.org/site/guide" target="_blank">http://gigadb.org/site/guide</a>YesAdditional CommentsIs the data acquisition clear, complete and methodologically sound?YesAdditional CommentsIs there sufficient detail in the methods and data-processing steps to allow reproduction?YesAdditional CommentsI think this is a one-off study. Is there sufficient data validation and statistical analyses of data quality? YesAdditional CommentsIs the validation suitable for this type of data?YesAdditional CommentsIs there sufficient information for others to reuse this dataset or integrate it with other data?YesAdditional CommentsTable 1 (Fields of the Google datasheet and their description) and unnumbered table (Taxa included): these are large tables and could be hosted in GigaDB in plain text. 
Any Additional Overall Comments to the AuthorThe "Taxa included" table needs to be numbered. RecommendationAccept

---

## [Reviewer Report]

Reviewer name and names of any other individual's who aided in reviewer Wim Van BortelDo you understand and agree to our policy of having open and named reviews, and having your review included with the published papers. (If no, please inform the editor that you cannot review this manuscript.)YesIs the language of sufficient quality?NoPlease add additional comments on language quality to clarify if needed
The authors should carefully check spelling errorsAre all data available and do they match the descriptions in the paper? YesAdditional CommentsAre the data and metadata consistent with relevant minimum information or reporting standards? See GigaDB checklists for examples <a href="http://gigadb.org/site/guide" target="_blank">http://gigadb.org/site/guide</a>YesAdditional CommentsIs the data acquisition clear, complete and methodologically sound?YesAdditional Commentssee some additional comments in my review regarding the description of what actually is part of the Bonne-Wepster collection Is there sufficient detail in the methods and data-processing steps to allow reproduction?YesAdditional CommentsIs there sufficient data validation and statistical analyses of data quality? YesAdditional Commentsnot relevant for this paperIs the validation suitable for this type of data?YesAdditional CommentsIs there sufficient information for others to reuse this dataset or integrate it with other data?NoAdditional CommentsI agree with the proposed possible re-use of the data. Yet, as the original taxonomic nomenclature is kept (which is very good). yet, the old taxonomic names should be ‘translated’ to the current taxonomy of the data is used a a current analysis. It would be good to state that explicitly and warn possible users of this dataset, or to provide the current taxonomic nomenclature in an additional field of the digitized database Any Additional Overall Comments to the AuthorThis data paper describes the digitizing of the Culicidae collection of Bonne-Wepster. This is an important work to make ‘old’ data available for current research. As the original taxonomic nomenclature is kept in the digitized database, the old taxonomic names should be ‘translated’ to the current taxonomy to make them useful for current and future scientific analysis. It would be good to state that explicitly and warn possible users of this dataset or to provide in the database the current taxonomic nomenclature in an additional field. Please find some more detailed comments below.

Data description
- It would be good to indicate when the mosquitoes were actually collected (e.g. after lines 33-34, and after lines 35-36).
- Line 38. When was the mosquito collection transferred to the Natural History Museum of Naturalis (NBC), in Leiden, The Netherlands?
- Lines 52-54. It is not clear whether the 1216 mosquitoes of the former Zoologische Museum Amsterdam Nederland (ZMAN) collection, belong to the Joahnna Bonne-Wepster collection; or is this a side project of the actual main project “Digitizing the Culicidae collection of Bonne-Wepster”. Please clarify this.

Context
- Lines 63-64. Change “for conservation purposes and biodiversity analysis” into “ to analyse biodiversity for example for conservation purposes”
- Line 69. Change diseases into pathogens. (see also line 95)
- Line 70. Please refer the most recent World Malaria Report of 2022.

Material and methods
- Line 124. What do you mean by “All drawers containing mosquitoes were sampled”. Did you only check a sample of all drawers or did you check and investigate all specimen? From the description (here and at other places in the text) it is not clear what you consider as former Bonne-Wepster collection (see also comment in the data description section). 
- Lines 139-142. It would be nice to add a figure with the new label that was added. 
- Lines 143-149. Likewise as (lines 139-142).
- Line 156 (data table). Row “Name comments” Write in full “det.”

Geographic coverage
- Line 206. Why is France between brackets?

Re-Use Potential
- I agree with the proposed possible re-use of the data. Yet, as the original taxonomic nomenclature is kept, the old taxonomic names should be ‘translated’ to the current taxonomy. It would be good to state that explicitly and warn possible users of this dataset, or to provide the current taxonomic nomenclature in an additional field of the digitized database .

Please carefully check the text for spelling errors (e.g., Line 92. Change “Durch” in “Dutch. Line 95. Check the spelling of mosquito and mosquitoes, etc).
RecommendationMinor Revision

---

## [Reviewer Report]

Reviewer name and names of any other individual's who aided in reviewer Cássio Lázaro Silva InacioDo you understand and agree to our policy of having open and named reviews, and having your review included with the published papers. (If no, please inform the editor that you cannot review this manuscript.)YesIs the language of sufficient quality?YesPlease add additional comments on language quality to clarify if needed
Are all data available and do they match the descriptions in the paper? NoAdditional CommentsUnfortunately, I was unable to find the data on GBIF.Are the data and metadata consistent with relevant minimum information or reporting standards? See GigaDB checklists for examples <a href="http://gigadb.org/site/guide" target="_blank">http://gigadb.org/site/guide</a>YesAdditional CommentsIs the data acquisition clear, complete and methodologically sound?YesAdditional CommentsIs there sufficient detail in the methods and data-processing steps to allow reproduction?YesAdditional CommentsIs there sufficient data validation and statistical analyses of data quality? YesAdditional CommentsIs the validation suitable for this type of data?YesAdditional CommentsIs there sufficient information for others to reuse this dataset or integrate it with other data?YesAdditional CommentsAny Additional Overall Comments to the AuthorThe text describes the digitization of data from the Bonne-Wepster Collection, which was collected over several decades from various regions around the world. Although the manuscript's title refers specifically to the Bonne-Wepster Collection, data from other collections, including those from the ZMAN and RMNH, were also digitized and included in the manuscript, as well as information from field notebooks.
The manuscript contains valuable scientific information about insects worldwide and has great potential for reuse. However, some parts of the manuscript require further attention and could benefit from revision during the review process.
In terms of context, it's important to note that while mosquitoes are often associated with disease transmission, they do not transmit diseases themselves. Rather, the agents that cause diseases, such as the plasmodia that cause malaria, are transmitted through mosquito bites.
Additionally, the authors should clarify whether the Bonne-Wepster, RMNH, and ZMAN collections are all part of a single large collection or if they are separate collections located in different places.
It would also be helpful if the authors could provide examples of photographs of the holotypes mentioned in the manuscript, which would help to clarify the content.
It would be helpful if the authors included the complete links for references 21 and 22.
Unfortunately, I was unable to find the data for references 21 and 22.
Finally, the authors need to ensure the accuracy of the taxonomic coverage presented in the manuscript. For example, some of the taxa listed, such as Chaoborus, Chironomus, Simulium, and Tipulida (?), do not belong to the Culicidae family and should be reviewed for accuracy.RecommendationMinor Revision